# Epithelioid Hemangioma of the Nose: A Challenging Diagnosis

**DOI:** 10.3390/healthcare10040633

**Published:** 2022-03-28

**Authors:** Fabrizio Schonauer, Annachiara Cavaliere, Giuseppe Pezone, Armando Calogero, Caterina Sagnelli, Antonello Sica, Luca D’Andrea, Antonello Baldo

**Affiliations:** 1Unit of Plastic Surgery, University of Naples Federico II, 80131 Naples, Italy; fschona@libero.it (F.S.); giupezone92@gmail.com (G.P.); 2Department of Advanced Biomedical Sciences, University of Naples Federico II, 80131 Naples, Italy; armando.calogero2@unina.it; 3Department of Mental Health and Public Medicine, Section of Infectious Diseases, University of Campania Luigi Vanvitelli, 80131 Naples, Italy; caterina.sagnelli@unicampania.it; 4Department of Precision Medicine, University of Campania Luigi Vanvitelli, 80131 Naples, Italy; antonello.sica@fastwebnet.it; 5Eye Clinic, Department of Neurosciences, Reproductive Sciences and Dentistry, University of Naples Federico II, 80131 Naples, Italy; dandrea.luca91@gmail.com; 6Department of Dermatology, University of Naples Federico II, 80131 Naples, Italy; baldo@unina.it

**Keywords:** epithelioid hemangioma, vasoproliferative disease, lesions of the nose

## Abstract

Epithelioid hemangioma is a rare reactive vasoproliferative disease presenting with painless vascular nodules in the dermal and subcutaneous tissues of the head and neck. Clinical diagnosis can be difficult as, in most cases, the only symptom is a progressively tender swelling next to a vessel course. Thus far, few cases of epithelioid hemangioma localized to the nose have been described in the literature. Herein, we present a case of a 47-year-old woman with just such a lesion of the nose, focusing on its diagnosis and treatment.

## 1. Introduction

Epithelioid hemangioma (EH, ISSVA classification 2018) [1] is a rare reactive vasoproliferative disease presenting with painless vascular nodules in the dermal and subcutaneous tissues of the head and neck. It is also known as angiolymphoid hyperplasia with eosinophilia, histiocytoid hemangioma, inflammatory angiomatous nodule, atypical granuloma and pseudopyogenic granuloma. Typically, EH arises in the head and neck region of young women, presenting as raised colored itchy nodules. While it rarely regresses spontaneously, there are no reports of malignant transformation [2,3].

As EH is quite rare, it can be clinically misdiagnosed for a number of benign inflammatory conditions such as Kimura disease, IgG4-related skin disease, bacillary angiomatosis, cutaneous epithelioid angiomatous nodule and malignant epithelial vascular tumors such as epithelioid sarcoma-like hemangioendothelioma, epithelioid hemangioendothelioma and epithelioid angiosarcoma [4,5]. EH can also be confused with more common conditions such as lipomas, sebaceous cysts and skin neoplasms such as basal and squamous cell carcinoma [6,7].

Although the etiology of EH remains unclear, surgical excision with negative margins has proven to be the treatment of choice.

Thus far, few cases of EH localized to the nose have been described, rather incompletely, in the literature, but this is the first case reported of a nasal tip involvement. Herein, we present in detail a case of a 47-year-old woman with just such a lesion of the nose, focusing on its diagnostic iter, emphasizing the importance of a radical surgical excision with a one-year follow-up experience.

## 2. Case Report

A 47-year-old woman was referred to our department with a solitary, painless but itchy nodule occupying the tip and part of the nasal dorsum. The lesion onset dated 2 years. The patient had no family history of any cutaneous diseases or vascular malformations and no history of trauma. A punch biopsy showed a rich amount of sebaceous adnexa, a periannesial and perivascular lymphoplasmacellular infiltrate with vascular ectasias and mild fibrosis, suggesting the possibility of discoid erythematous lupus.

The patient started topical treatment but with no benefits. The patient’s lesion progressively increased in size and after one year, on examination, it had developed into a fixed lump covered by thin whitish skin with small telangiectasias (Figure 1).

A Doppler ultrasonography using an 18 Mhz linear transducer revealed a highly vascularized nodular subcutaneous lesion of 23 mm × 16 mm (Figure 2).

A biochemistry test, routine hematology and erythrocyte sedimentation rate were all within the normal reference ranges.

While refusing another biopsy, the patient requested the total excision of the lesion and, therefore, a definitive diagnosis was postponed until formal lesion excision.

Subsequently, an excision under local anesthesia with sedation was performed. The residual defect resulting from the wide excision was covered using a full-thickness skin graft harvested from the preauricular region (Figure 3).

At histology, hematoxylin and eosin-stained sections showed redundant vascular proliferation with multilobular architecture occupying the medium and deep dermis consisting of small caliper vessels with mildly hyperplastic endothelium and epithelioid aspects (Figure 4). A rich lymphocytic infiltration with a predominant eosinophilic quote was associated. An immunohistochemistry stain showed the presence of the vascular proliferation markers CD31+, CD34+ and ERG1+. In accordance with these findings, the lesion was diagnosed as epithelioid hemangioma.

At the one-year follow-up, the patient had no sign of recurrence and was satisfied with the result (Figure 5). A further nasal tip surgical revision was offered but she refused.

## 3. Discussion

EH is an uncommon benign vascular proliferative lesion of unknown origin with a good prognosis and no reported incidence of metastatic disease. Nevertheless, these lesions have high rates of local recurrence and in some cases can have a simultaneous multifocal presentation, even in different organs [6]. Some authors suggest that traumatic events may trigger the onset of EH [8]. EH is commonly found in the small vessels of the dermis or subcutis, but it can also affect larger muscular arteries such as the facial artery and the temporal artery [4].

Clinical diagnosis can be difficult as, in most cases, the only symptom is a progressively tender swelling next to a vessel course. Skin discoloration may take place, especially in districts with a terminal circulation such as the nose [8]. Pain and/or functional symptoms may rarely manifest because of tumor compression on adjacent structures (nerves, vessels or tendons). Imaging can be useful for studying tumor extension, vascular involvement and any other simultaneous presentations, but a histology is mandatory in order to obtain a definitive diagnosis. Microscopic features include lined vessels with atypical endothelial cells which appear plump/swollen and epithelioid, associated with an eosinophilic and lymphocytic infiltration in the perivascular tissue. In 20% of the cases, generalized eosinophilia can be present and, occasionally, accompanied by raised serum IgE levels [4,6]. Endothelial cell markers such as CD31 or CD34 can confirm diagnosis [9].

The differential diagnosis of EH should include Kimura disease, pyogenic granuloma and Kaposi sarcoma [10].

Surgical excision is the treatment of choice for EH. Due to its high risk of recurrence, a wide radical excision should be performed [11]. When important facial areas with a high aesthetic impact such as the nose are involved, reconstruction can be difficult. Depending on the defect size left after a wide excision, different options can be considered [12,13,14]. In this patient, because the punch biopsy result was not diagnostic and due to the uncertain nature of the lesion, we preferred to cover the defect with a full thickness skin graft. A local flap-based reconstruction was reserved if any further excision widening was necessary.

Once completely removed, EH local recurrence is rare; however, with an incomplete resection, almost 33% of patients have a recurrence, either at the same site or a distant site along the course of the blood vessel from where the EH originated [5,11].

The nasal localization of EH is uncommon and only six cases have been reported in the literature so far [10,15,16,17,18,19]. Cutaneous involvement has been reported in only two cases [10,17]. Surgical excision has been the treatment of choice in all cases, but no details on reconstruction techniques or possible complications have been provided. Furthermore, no follow-up data have been reported, with the exception of Sedran et al who followed a patient for seven years and Youssef et al. who reported a lesion recurrence, which was treated with intralesional prednisolone and a 0.1% Tacrolimus ointment.

To conclude, HE groupings include a variety of rare vascular lesions that can affect different tissues and organs. Diagnostic investigations should include imaging techniques to determine the extent of the lesion and to confirm vascular involvement. Once the diagnosis of HE is suspected, an early and complete surgical excision of the lesions with a thorough histological and immunohistochemical analysis should be undertaken.

## Figures and Tables

**Figure 1 healthcare-10-00633-f001:**
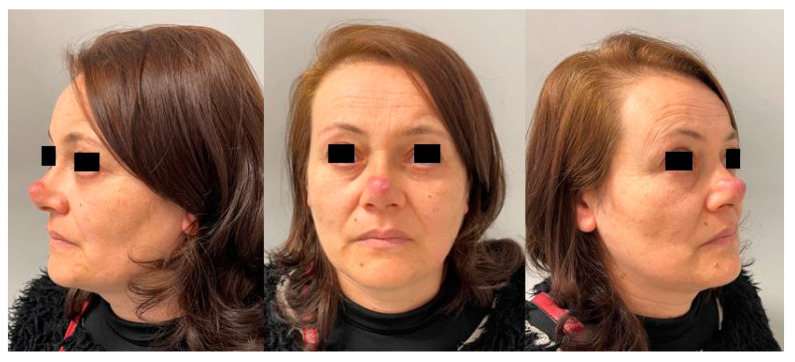
Preoperative appearance of the lesion.

**Figure 2 healthcare-10-00633-f002:**
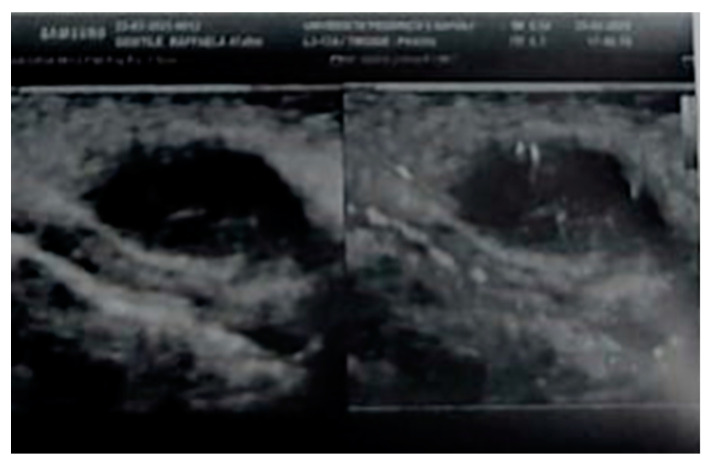
Doppler ultrasonography of the lesion.

**Figure 3 healthcare-10-00633-f003:**
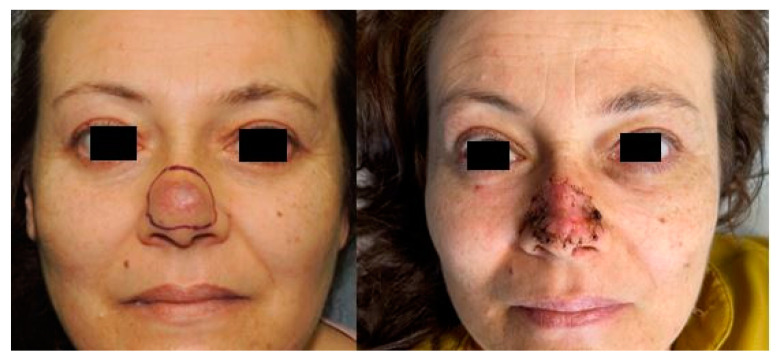
Preoperative planning of wide excision of the lesion (**left**); 10-day postoperative result (**right**).

**Figure 4 healthcare-10-00633-f004:**
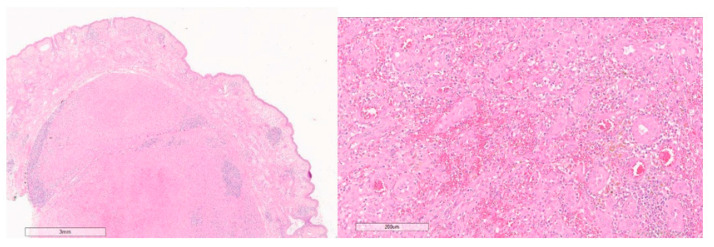
Histopathologic study: pseudoepitheliomatous hyperplasia with rich lymphocytic infiltration with predominant eosinophilic quote in fibrous stroma (**left**), vascular proliferation with multilobular architecture with prominent endothelial cells (**right**); hematoxylin and eosin.

**Figure 5 healthcare-10-00633-f005:**
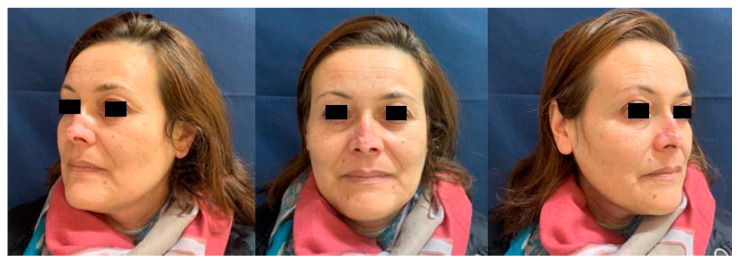
One-year follow-up, no sign of recurrence.

## Data Availability

Not applicable.

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
