# Peer review of "Epithelioid Hemangioma of the Nose: A Challenging Diagnosis"

_healthcare, 2022, doi:10.3390/healthcare10040633_

Round 1
Reviewer 1 Report
This manuscript is a well-documented case report regarding the case of Histiocytoid Hemangioma of the nose. The clinical course is beautifully presented, and the discussion is satisfactory.
I would like to request two improvements.
1)Because the tumor is very rare, the individual patient may be identified. If you don't think it's essential information, it's safe to delete the actual year and month (ex. 2021 April).
2)In the discussion part, the authors have introduced some skin flaps as reconstructive procedures. However, in this case, the authors chose to reconstruct by skin grafting. It would be more convincing for the reader to describe in some more detail the reason for choosing skin grafting in this case and its superiority over flaps.
Author Response
I would like to request two improvements.
1) Because the tumor is very rare, the individual patient may be identified. If you don't think it's essential information, it's safe to delete the actual year and month (ex. 2021 April).
We’ve deleted any temporal information that could help identify the patient.
2) In the discussion part, the authors have introduced some skin flaps as reconstructive procedures. However, in this case, the authors chose to reconstruct by skin grafting. It would be more convincing for the reader to describe in some more detail the reason for choosing skin grafting in this case and its superiority over flaps.
“…In this patient, because the punch biopsy result was not diagnostic and due to the uncertain nature of the lesion, we preferred to cover the defect with a full thickness skin graft. A local flap based reconstruction was reserved if any further excision widening was necessary…” This explanation has been added to the manuscript.
Reviewer 2 Report
Dear Authors,
Here I recommend some suggestions to improve the quality of your article.
- To avoid confusion with multiple names of the same lesion, please use the official name of the lesion, according to the ISSVA classification.
- The image with Doppler ultrasonography of the lesion is unclear,
please provide a better one. - The images with histologic diagnosis should be provided
- The Discussion section should include the presented case compared to other similar cases.
Author Response
To avoid confusion with multiple names of the same lesion, please use the official name of the lesion, according to the ISSVA classification.
The lesion name has been changed accordingly to ISSVA classification in the manuscript. ISSVA classification has been cited.
The image with Doppler ultrasonography of the lesion is unclear, please provide a better one.
Unfortunately, no other ultrasonography image is available.
The images with histologic diagnosis should be provided
Two images with the histologic diagnosis have been added to the manuscript.
The Discussion section should include the presented case compared to other similar cases.
“…Nasal localization of EH is uncommon and only seven cases have been reported in literature so far, but none involving the nasal tip. As for the other cases reported, the initial diagnosis has been difficult and only the histological examination revealed the nature of the lesion...” Comparison to other similar cases has been added to the manuscript and citations have been provided.
Reviewer 3 Report
Thanks for submitting this interesting case, which is a rare diagnosis especially on the nose. This case report is well-written and presents the case clearly along with an adequate background. I would suggest mentioning the donor site for the full-thickness skin graft.
Author Response
I would suggest mentioning the donor site for the full-thickness skin graft.
“…The residual defect resulting from the wide excision was covered using a full thickness skin graft harvested from the preauricular region...” The donor site of the skin graft has been mentioned in the manuscript
Round 2
Reviewer 2 Report
Dear authors,
I consider that the manuscript has been improved, but it still needs to be completed.
After the Introduction, I think it would be useful to introduce a mini-subchapter with the Justification of the case presentation: why is this case special and why it is worth presenting.
I also consider that in the Discussions chapter it would be useful to compare the case presented here with the other seven cases in the literature. How is your case different from other cases besides location? What was the treatment method chosen for the other cases, what complications can occur, what was the long-term evolution, etc.
Is there approval from the local ethics committee to publish this case?
Author Response
Dear authors,
I consider that the manuscript has been improved, but it still needs to be completed.
After the introduction, I think it would be useful to introduce a mini-subchapter with the Justification of the case presentation: why is this case special and why it is worth presenting.
“..Thus far, few cases of EH localized to the nose have been described, rather incompletely, in the literature, but this is the first case reported of nasal tip involvement. Herein we present in detail a case of a 47-year-old woman with just such a lesion of the nose, focusing on its diagnostic iter, emphasizing the importance of a radical surgical excision with a one year follow up experience..”
The introduction has been modified as suggested. Previously reported cases lacked details, especially concerning surgical treatment and follow up. Since the EH is rare, the more detailed is a case report the better is to help other practitioners with diagnosis and treatment.
I also consider that in the Discussions chapter it would be useful to compare the case presented here with the other seven cases in the literature. How is your case different from other cases besides location? What was the treatment method chosen for the other cases, what complications can occur, what was the long-term evolution, etc.
“..Nasal localization of EH is uncommon and only six cases have been reported in the literature so far. [10, 15-19] Cutaneous involvement has been reported in only two cases. [10, 17] Surgical excision has been the treatment of choice in all cases, but no details on reconstruction techniques or possible complications have been provided. Furthermore, no follow up data have been reported, with the exception of Sedran et al, who followed a patient for seven years, and Youssef et al. who reported lesion recurrence, which was treated with intralesional prednisolone and 0,1% Tacrolimus ointment..”
A comparison with the previously published papers has been added to the discussion chapter. Please note that the cases reported are six, instead of seven. A case report paper was mistakenly added.
Is there approval from the local ethics committee to publish this case?
There was no need for local ethics committee approval to publish this case.